# Changes in dynamic transitions between integrated and segregated states underlie visual hallucinations in Parkinson's disease

Angeliki Zarkali [1✉], Andrea I. Luppi [2,3], Emmanuel A. Stamatakis [2,3], Suzanne Reeves[4], Peter McColgan[5], Louise-Ann Leyland[1], Andrew J. Lees[6] & Rimona S. Weil[1,7,8]

Hallucinations are a core feature of psychosis and common in Parkinson's. Their transient, unexpected nature suggests a change in dynamic brain states, but underlying causes are unknown. Here, we examine temporal dynamics and underlying structural connectivity in Parkinson's-hallucinations using a combination of functional and structural MRI, network control theory, neurotransmitter density and genetic analyses. We show that Parkinson's-hallucinators spent more time in a predominantly Segregated functional state with fewer between-state transitions. The transition from integrated-to-segregated state had lower energy cost in Parkinson's-hallucinators; and was therefore potentially preferable. The regional energy needed for this transition was correlated with regional neurotransmitter density and gene expression for serotoninergic, GABAergic, noradrenergic and cholinergic, but not dopaminergic, receptors. We show how the combination of neurochemistry and brain structure jointly shape functional brain dynamics leading to hallucinations and highlight potential therapeutic targets by linking these changes to neurotransmitter systems involved in early sensory and complex visual processing.

[1] Dementia Research Centre, University College London, 8-11 Queen Square, London WC1N 3AR, UK. [2] Division of Anaesthesia, School of Clinical Medicine, University of Cambridge, Cambridge CB2 0QQ, UK. [3] Department of Clinical Neurosciences, University of Cambridge, Cambridge CB2 0QQ, UK. [4] Division of Psychiatry, University College London, 149 Tottenham Court Rd, London W1T 7BN, UK. [5] Huntington's Disease Centre, University College London, Russell Square House, London WC1B 5EH, UK. [6] Reta Lila Weston Institute of Neurological Studies, University College London, 1 Wakefield Street, London WC1N 1PJ, UK. [7] Wellcome Centre for Human Neuroimaging, University College London, 12 Queen Square, London WC1N 3AR, UK. [8] Movement Disorders Consortium, University College London, London WC1N 3BG, UK. ✉email: a.zarkali@ucl.ac.uk

Psychotic disorders cause significant global burden to affected individuals, families and healthcare systems. In Parkinson's disease (PD), psychosis is common, and visual hallucinations are associated with cognitive decline[1], poorer quality of life[2] and increased mortality[3]. However, despite their impact, the brain changes that give rise to psychotic hallucinations are not fully understood. The transient, unpredictable nature of hallucinations, even in patients who regularly experience them, suggests they relate to changes in dynamic brain processes and shifts in states. Resting-state functional MRI (rsfMRI) measures spontaneous fluctuations in brain activity based on correlated fluctuations in blood oxygenation[4] and has shown changes in the relative activity of specific functional brain networks in patients with PD-hallucinations[5], with increased activation of the default mode network (DMN) and impaired recruitment of the dorsal attention network[6–9]. However, these studies only provide a static image of functional connectivity, calculated over an entire scanning period, rather than examining dynamic changes in brain states.

An extension of this approach is dynamic functional connectivity analysis, which measures spontaneous fluctuations in connectivity over time[10–12] and may be a more accurate representation of fluctuating cognitive states than previous static approaches[13]. Changes in temporal dynamics are seen in schizophrenia and other psychiatric conditions[14–17], and recent work showed imbalance of temporal dynamics of integrated and segregated states in anaesthesia and disorders of consciousness[18,19] and after administration of the psychedelic LSD, known for its hallucinogenic properties[20]. Changes in dynamic functional connectivity are described in PD[21] and are associated with severity of both motor and cognitive symptoms[22–24] but are as yet unexplored in relation to neuropsychiatric symptoms.

Functional connectivity is likely to be affected by breakdown in the anatomical connections between regions. Indeed, PD patients with hallucinations show widespread disruption in structural connections between brain regions, measured using diffusion MRI[25,26]. These changes particularly affect highly connected brain regions or "hubs" important for switching the brain between different states[27,28]. Network control theory is a mathematical framework developed to study how the activity of a network's nodes is influenced by the network's structure. In the context of neuroscience, it offers a mechanistic explanation of how the brain transitions between cognitive states based on its structure, enabling behaviour[29]. It integrates information from an individual's structural connectome (white matter connectivity derived from diffusion-weighted imaging) and temporal activation patterns (derived for example from fMRI) to specify how observed temporal activation patterns are constrained by the structural connectome[29,30]. This framework defines brain states as the magnitude of haemodynamic activity across brain regions at a single time point and assumes that the brain's activation state at a given time is a linear function of a previous state, the underlying structural connectome and the additional control energy that is added to the system[29,31]. In this way, the minimal energy cost needed to move the brain from one state to another can be calculated based on its structural network[29,31,32]. A state that is less energy-demanding to maintain, or requires lower energy for transition, will be preferred. Recent work has shown that certain state transitions are preferable in the resting brain over others but this can be overcome by cognitive demands and is related to brain development and cognition[33–36]. This framework has the potential to explain why a particular state is predominantly seen in health and how the balance between states may change in the presence of disease.

Transitions between functional states may be modulated by neurotransmitter systems[37]. Dopamine transmission, particularly D2 receptor expression guides state transitions during a working memory task[34]. Excess dopamine release is a core neurobiological theory of schizophrenia[38], and excess striatal dopamine has been linked to hallucinatory experiences[39]. Dopamine has long been considered the key driving neurotransmitter for PD hallucinations[40] with higher daily levodopa doses associated with higher risk of hallucinations[41,42]. However, recent studies have challenged this model and implicated other neurotransmitters in PD-hallucinations: higher density of 5HT2A serotonin receptors[43], reduced GABA concentration[44] and cholinergic neuronal loss[45] have each been described in patients with PD and visual hallucinations. The role of dopamine in cognitive state transitions in health has also been challenged, with regional expression patterns of inhibitory and facilitatory neurotransmitters other than dopamine recently linked to dynamic functional states[46] and both noradrenaline[12,47] and serotonin[37] driving whole-brain functional connectivity changes. A better understanding of the complex changes in neurotransmitter systems causing hallucinations would inform the development of more effective and targeted treatments for this distressing symptom.

Here, we aimed to investigate the nature of temporal dynamics in PD-associated visual hallucinations using rsfMRI; and determine whether the balance between predominantly Integrated and *Segregated* states of functional connectivity is altered in PD patients with hallucinations compared to patients without hallucinations and controls (overview in Fig. 1). We found that PD patients with hallucinations show impaired temporal dynamics, with a predisposition towards a predominantly Segregated state of functional connectivity. We then applied network control theory to calculate each individual's required energy cost to transition from the integrated-to-the-segregated state and vice versa, and the cost to maintain each state. We found that Parkinson's-hallucinators required less energy to transition from the integrated-to-segregated state than those without hallucinations and controls. Finally, we identified the brain regions that contribute most to the transition from integrated-to-segregated state. As dynamic neural systems are modulated by neurotransmitter systems[37,47] we related the spatial organisation of this transition to regional neurotransmitter distribution using PET-derived density profiles and regional gene expression for neurotransmitter receptors.

## Results

Ninety-one patients with PD were included: 16 PD patients with habitual visual hallucinations (PD-VH), 75 PD patients without hallucinations (PD-non-VH) and 32 controls. Demographics and clinical assessments are seen in Table 1. All participants experienced hallucinations in the visual domain, with details on the experienced hallucinatory images in Table 2. PD-VH and PD-non-VH were well matched in demographics, cognitive and motor performance, levodopa equivalent dose, and image quality and motion parameters (Table 1 and Supplementary Table 1). PD-VH participants showed higher depression scores ($p = 0.032$) than PD-non-VH participants but well below the clinical threshold for depression ($\geq 8$). As a result of the presence of hallucinations and higher depression burden ($p = 0.014$), PD-VH participants showed higher total UPDRS scores, which assessed non-motor symptoms, but they did not differ in terms of motor severity or levodopa equivalent dose. Although disease duration differed between PD-VH and PD-non-VH participants ($p = 0.044$), there was no correlation between disease duration and temporal functional changes ($\rho = -0.110$, $p = 0.297$ between proportion of time spent in an *Integrated* vs *Segregated* state and disease duration in PD participants) therefore we did not correct for disease duration in our main comparisons of interest.

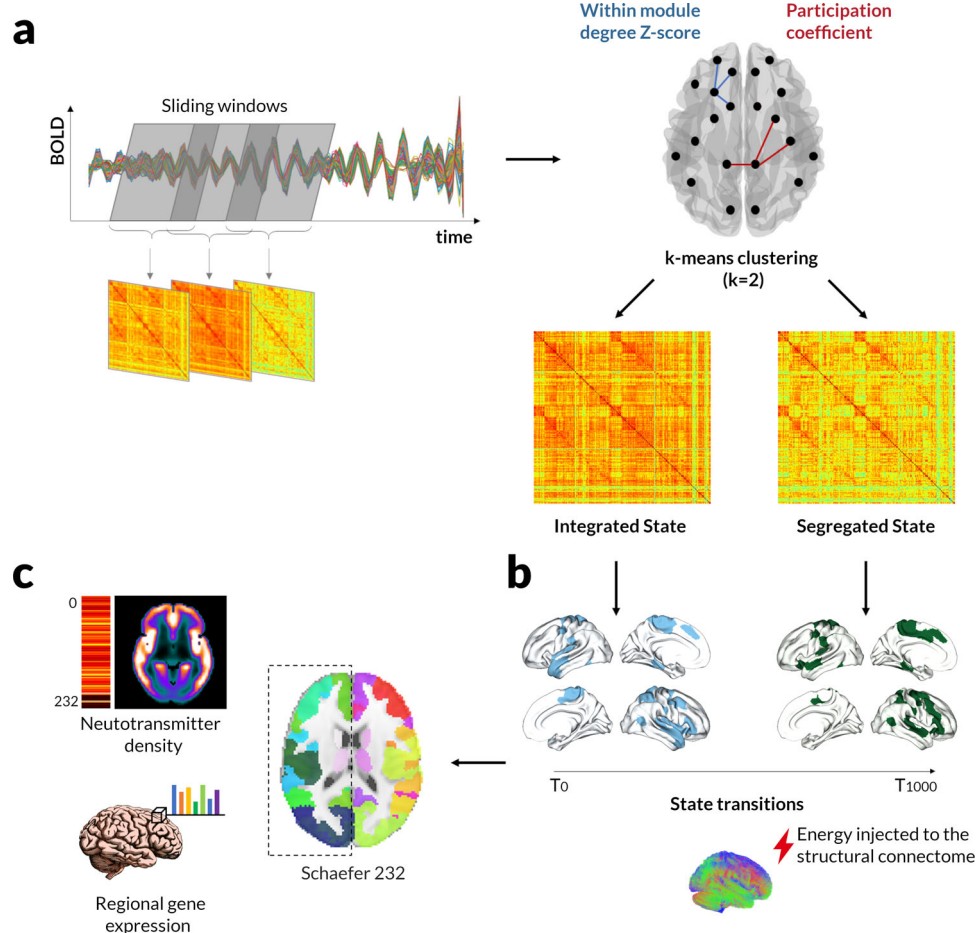

**Fig. 1 Overview of the study methodology. a** Deriving *Integrated* and *Segregated* states of dynamic functional connectivity. After obtaining sliding-windows (each 44 s duration) of dynamic functional connectivity for each participant, the joint histogram of participation coefficient and within-module degree Z-score was used for *k*-means clustering (*k* = 2) (BOLD, blood oxygen level dependent activity). The cluster with highest average participation coefficient is then identified as the predominantly *Integrated* dynamic state and the cluster with the lowest participation coefficient as the predominantly *Segregated* state. Note that this is done for each participant separately leading to individually-defined states. **b** Modelling state transitions. After deriving each individual's *Integrated* and *Segregated* states we used an optical control framework to calculate the minimal control energy that needs to be applied to each node of the structural network to transition from a baseline state at time $T_0$ to a target state at time $T_{1000}$. Here, as an example, we illustrate the transition from the *Integrated* state (top 20% of nodes in blue) to the *Segregated* state (top 20% of nodes in green) but minimal energies were also calculated for segregated-to-integrated transition as well as minimal energies to maintain the *Integrated* state (integrated-to-integrated) and *Segregated* state (segregated-to-segregated) using the same model. Minimal control energies were calculated for each subject based on their structural brain network, which was estimated using diffusion imaging and probabilistic tractography. Both states were represented in the model as a vector of the sum connectivity strength for each node (1*232). **c** Linking with neurotransmitter systems. Minimal control energies to transition between and maintain functional states were compared between patients with PD with (PD-VH, *n* = 16) and without hallucinations (PD-non-VH, *n* = 75). Transitions that differed between groups were then further explored to examine whether contributing nodes (requiring mode control energy) were associated with specific neurotransmitter systems. To do this, we calculated for each of the 232 regions of interest of our parcellation (Schaeffer 232: 200 cortical and 32 subcortical regions) (1) mean neurotransmitter density profiles derived from PET data (serotonin (*5HT1a, 5HT2a* and *5HT1b*), dopamine (*D1* and *D2*) and *GABA$_A$* receptors) and (2) gene expression profiles for each of 31 pre-selected genes encoding receptors for norepinephrine, acetylcholine, dopamine and serotonin.

**Preserved topology of functional connectivity states**. To examine the dynamic changes in functional connectivity underlying PD-hallucinations, we employed an a priori clustering of dynamic functional connectivity into two states of functional connectivity, an *Integrated* and a *Segregated* state. After obtaining sliding-windows (44 s duration each) of dynamic functional connectivity for each participant, the joint histogram of participation coefficient and within-module degree Z-score for *k*-means clustering (*k* = 2; independently confirmed as optimal number of clusters on data-driven evaluation, Supplementary Fig. 1). The cluster with highest average participation coefficient was identified as the *Integrated* dynamic state and the cluster with the lowest participation coefficient as the S*egregated* state, as

previously described[12,18,20,48,49]. This was performed separately for each participant (using the same criteria) leading to individually-defined *predominantly Integrated* and *Segregated* states (Fig. 1a). Differences between the two states are seen in Supplementary Fig. 2 and Supplementary Table 3.

These two states did not significantly differ between groups (PD versus controls or PD-VH versus PD-non-VH) when comparing connectivity strength in each state using network-based statistics, or between-group differences in density (*Integrated*: Kruskal–Wallis $H = 2.473$, $p = 0.290$, *Segregated*: 0.175, $p = 0.529$), entropy of connectivity values (*Integrated*: $H = 0.723$, $p = 0.696$, predominantly-*Segregated*: $H = 0.905$, $p = 0.636$), structural-functional coupling (*Integrated* $F(111,2) = 1.093$,

**Table 1 Demographics and clinical assessments in patients with Parkinson's with hallucinations (PD-VH) and without hallucinations (PD-non-VH).**

| Attribute | | Controls $n = 32$ | PD-non-VH $n = 75$ | PD-VH $n = 16$ | *p* value |
|---|---|---|---|---|---|
| Demographics | Age (years) | 66.1 (9.4) | 64.4 (7.8) | 64.8 (8.6) | 0.653 |
| | Male (%) | 13 (40.6) | 41 (54.7) | 5 (31.2) | 0.029[a] |
| | Years in education | 17.8 (2.5) | 16.9 (2.6) | 17.5 (3.6) | 0.279 |
| | Total intracranial volume (ml) | 1390.7 (96.6) | 1479.0 (132.6) | 1407.3 (114.8) | 0.002[a] |
| Mood (HADS) | Depression score | **1.7 (1.9)** | **3.9 (3.0)** | **4.7 (3.4)** | **0.032[a,b,c]** |
| | Anxiety score | 4.0 (3.5) | 5.6 (3.8) | 7.0 (4.4) | <0.001[a] |
| Vision | Visual acuity (LogMAR)* | −0.08 (0.23) | −0.08 (0.16) | −0.07 | 0.351 |
| | Contrast sensitivity (Pelli Robson)* | 1.78 (0.2) | 1.79 (0.2) | 1.70 (0.2) | 0.106 |
| | Colour vision (D15 total error score) | 2.4 (6.9) | 3.4 (8.7) | 2.7 (4.6) | 0.681 |
| Cognition | MMSE | 29.0 (1.0) | 28.9 (1.2) | 28.6 (1.9) | 0.883 |
| | MOCA | 29.0 (1.3) | 28.2 (2.1) | 26.9 (3.4) | 0.047[a] |
| *Attention* | Digit span backwards | 7.2 (2.1) | 7.1 (2.3) | 7.9 (2.3) | 0.601 |
| | Stroop: colour (sec) | 32.1 (6.7) | 33.5 (7.6) | 38.1 (9.1) | 0.089 |
| *Executive function* | Stroop: interference (sec) | 55.4 (11.6) | 60.2 (19.2) | 69.6 (23.9) | 0.051 |
| | Category fluency | 22.5 (5.1) | 21.7 (5.9) | 19.8 (7.4) | 0.339 |
| *Memory* | Word Recognition Task | 24.3 (1.2) | 24.3 (1.2) | 23.8 (0.9) | 0.056 |
| | Logical Memory | 14.1 (4.1) | 13.5 (4.3) | 12.5 (4.6) | 0.617 |
| *Language* | Graded Naming Task | 22.5 (6.2) | 23.9 (2.9) | 23.7 (2.3) | 0.802 |
| | Letter fluency | 16.4 (5.4) | 16.7 (5.4) | 17.7 (5.3) | 0.509 |
| *Visuospatial* | Benton's Judgement of Line Orientation | 24.9 (5.6) | 24.5 (3.7) | 23.1 (5.3) | 0.338 |
| | Hooper | 25.7 (2.1) | 24.7 (2.8) | 23.3 (4.3) | 0.074 |
| Disease-specific measures | Disease duration | – | **3.9 (2.3)** | **5.3 (3.4)** | **0.044** |
| | UPDRS total score | – | **42.7 (20.8)** | **62.1 (38.5)** | **0.014** |
| | UPDRS part 3 (motor) | – | 21.2 (11.3) | 29.8 (22.6) | 0.129 |
| | UM-PDHQ (hallucination severity score) | – | – | 4.6 (2.4) | – |
| | LEDD (mg) | – | 437.0 (255.1) | 450.0 (221.2) | 0.295 |
| | RBDSQ | – | 4.0 (2.4) | 5.1 (2.5) | 0.055 |

All data shown are mean (SD) except gender.
In bold characteristics that significantly differed between the PD-VH and PD-non-VH.
[a]Significant difference between PD-VH and controls.
[b]Significant difference between PD-non-VH and controls.
[c]Significant difference between PD-VH and PD-non-VH.
*Best binocular score used; LogMAR: lower score implies better performance, Pelli Robson: higher score implies better performance. HADS: Hospital anxiety and depression scale; MMSE: Mini-mental state examination; MOCA: Montreal cognitive assessment; UPDRS: Unified Parkinson's disease rating scale; UM-PDHQ: University of Miami Hallucination Questionnaire (max score: 14); LEDD: Total Levodopa equivalent dose; RBDSQ: REM sleep behaviour disorder screening questionnaire.

---

**Table 2 Characteristics of visual hallucinations experienced by patients with Parkinson's disease (PD-VH).**

| Visual hallucinations characteristics | | PD-VH ($n = 16$) |
|---|---|---|
| Phenotype | Complex visual hallucinations | 11 (62.5%) |
| | Minor visual hallucinations | 5 (31.3%) |
| Frequency | Less than once a week[a] | 11 (62.5%) |
| | More than once a week | 5 (31.3%) |
| Duration | Less than 1 s | 8 (50.0%) |
| | Less than 10 s | 6 (37.5%) |
| | More than 10 s | 2 (12.5%) |
| Insight | Always preserved | 10 (62.5%) |
| | Sometimes preserved | 4 (25.0%) |
| | No insight | 2 (12.5%) |
| Number of experienced images mean (sd) | | 1.44 (0.79) |
| Distress | No distress | 10 (62.5%) |
| | Mild to moderate distress | 6 (37.5%) |

Participants were asked to reflect on all visual hallucinatory phenomena experienced within the previous month.
Complex visual hallucinations included well-formed imagery (people, animals, etc.), stationary or animate images. Minor hallucinations included passage hallucinations and non-formed images (shadows, etc.). Misperceptions alone were not included as minor hallucinations.
[a]All participants experienced hallucinations more frequently than once per month.

---

$p = 0.339$, *Segregated*: F(111,2) = 1.401, $p = 0.251$) or small world propensity (*Integrated*: $H = 1.065$, $p = 0.587$, *Segregated*: $H = 4.400$, $p = 0.111$).

**Impaired temporal properties of dynamic functional connectivity in patients with hallucinations**. Although the states themselves did not differ between groups, we found significant changes in their temporal properties. PD-VH spent a significantly smaller proportion of time in the *Integrated* state (therefore more time in the *Segregated* state) than PD-non-VH ($\beta = -0.113$, $p = 0.032$) and controls ($\beta = -0.128$, $p = 0.026$) (Fig. 2a). Within PD patients, the proportion of time spent in the *Integrated* state was inversely correlated with hallucination severity (Spearman's $\rho = -0.259$, $p = 0.013$). Mean dwell time (number of consecutive windows spent in each state) in the *Segregated* state was higher in PD-VH than PD-non-VH ($19.1 \pm 16.9$ in PD-VH vs $9.5 \pm 9.1$ in PD-non-VH $H = 4.058$, $p = 0.044$), but did not differ for the *Integrated* state ($H = 2.166$, $p = 0.141$). No differences were seen in dwell times of either state between PD and controls. Finally, the total number of transitions was lower in PD-VH than PD-non-VH ($5.7 \pm 5.3$ in PD-VH vs $8.5 \pm 6.2$ in PD-non-VH, $H = 3.87$, $p = 0.049$) (Fig. 2b). Results were replicated using a finer parcellation (Supplementary Fig. 4). Overall, this suggests

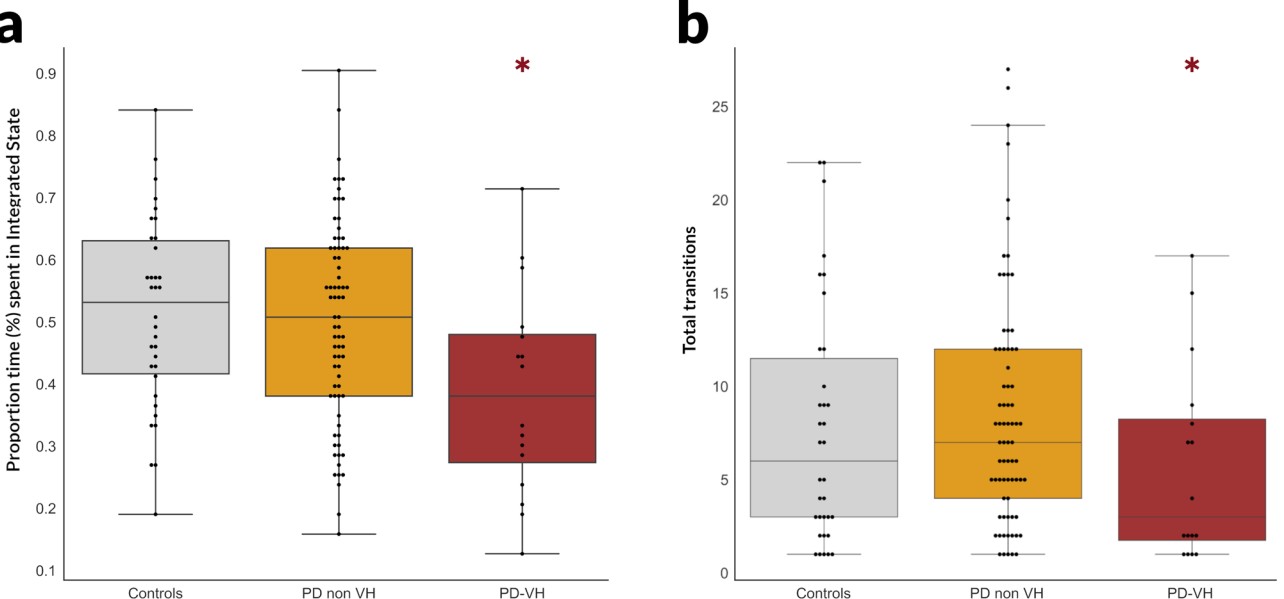

**Fig. 2 Altered temporal properties of dynamic functional connectivity in patients with Parkinson's and visual hallucinations. a** Percentage of total time spent in the *Integrated* state. Patients with Parkinson's with visual hallucinations ($n = 16$) spent significantly less time in the *Integrated* state of dynamic functional connectivity than patients without hallucinations ($n = 75$) ($p = 0.032$) and controls ($n = 32$) ($p = 0.0262$) (error bars are 95% confidence intervals). **b** Total number of transitions between states. Patients with Parkinson's and hallucinations ($n = 16$) had reduced overall transitions between states than patients without hallucinations ($p = 0.049$) (error bars are 95% confidence intervals). PD-VH: Parkinson's disease with visual hallucinations, PD-non-VH: Parkinson's disease without hallucinations.

that PD-VH spend more time in the *Segregated* state than PD-non-VH, with fewer total transitions and longer dwelling time within the *Integrated* state.

**Reduced energy costs to transition from the integrated to segregated state in patients with visual hallucinations.** Having identified significant differences in terms of brain dynamics between PD-VH and PD-non-VH, which are specifically related to the severity of visual hallucinations (the focus of our present investigation), we sought to interrogate further this difference between PD patients. We have previously shown widespread structural connectivity changes in PD-VH[28]. Given neural dynamics are constrained by the structural connectome, we used the framework of network control theory to integrate information about structural network topology and functional brain dynamics[29,31,32]. Using this framework, the minimal energetic cost to transition from one specific functional brain state (defined as the magnitude of brain activation at a specific time point) to another can be calculated using the structural brain network topology[31,32,50]; lower energetic costs required to transition to a specific state may make this transition preferable.

Specifically, we aimed to investigate whether the *Segregated* state predominance observed in hallucinators could be explained by differences in ease of transition from the integrated-to-segregated state or vice versa or a difference in ease of maintaining the *Segregated* state. To do this, we calculated the minimal control energy that needs to be applied to the structural network of each participant to (1) transition from integrated-to-segregated state, (2) transition from segregated-to-integrated state, (3) maintain the *Integrated* state and (4) maintain the *Segregated* state (Fig. 1b). Minimal control energies were calculated for each subject based on their structural brain network, which was estimated using diffusion imaging and probabilistic tractography. For the purposes of this calculation, and in contrast to previous publications, we represented the *Integrated* and *Segregated* states as a vector of sum functional

connectivity for each brain region. We then examined whether transition and persistence energies in each state differed between PD-VH and PD-non-VH.

Similarly to previous work in healthy individuals[34], persistence energy for the more connected *Integrated* state was higher than the *Segregated* state for all participants (repeated measures ANOVA main effect of *Integrated* to *Segregated* state persistence energy $F(1,113) = 12.432$, $p < 0.001$). Similarly the minimal energy needed to transition from the *Segregated* to *Integrated* state was higher ($F(1,113) = 6.722$, $p = 0.011$) (Supplementary Fig. 3). When we examined differences between patients with PD with and without hallucinations, PD-VH needed significantly lower control energy to transition from the *Integrated*-to-*Segregated* state than PD-non-VH (effect size Hedge's $g = 0.922$, $t = 2.376$, $p = 0.029$) (Fig. 3a). There were no statistically significant differences between PD-VH and PD-non-VH in the minimal control energy needed to transition from *Segregated*-to-*Integrated* state ($t = 1.346$, $p = 0.195$), or to persist within the *Integrated* ($t = 1.041$, $p = 0.312$) or *Segregated* state ($t = 1.079$, $p = 0.295$). Therefore, network control theory reveals that the higher proportion of time that PD-VH patients spend in the *Segregated* state may be accounted for in terms of this state being easier to transition to from the *Integrated* state (as opposed to being easier to persist in).

**Transition from integrated to the segregated state is driven by subcortical and more multimodal brain regions.** A further benefit of applying control theory to functional brain states is that it provides regional information about the cost of maintaining and transitioning between these states.

We therefore aimed to identify which brain regions contribute more to this transition from the *Integrated*-to-*Segregated* state (which nodes require more energy in order to transition, with high contributors defined as the top 20% of regions). These higher contributors are more likely to be responsible for the changes in energy costs seen in PD-VH (significantly less control

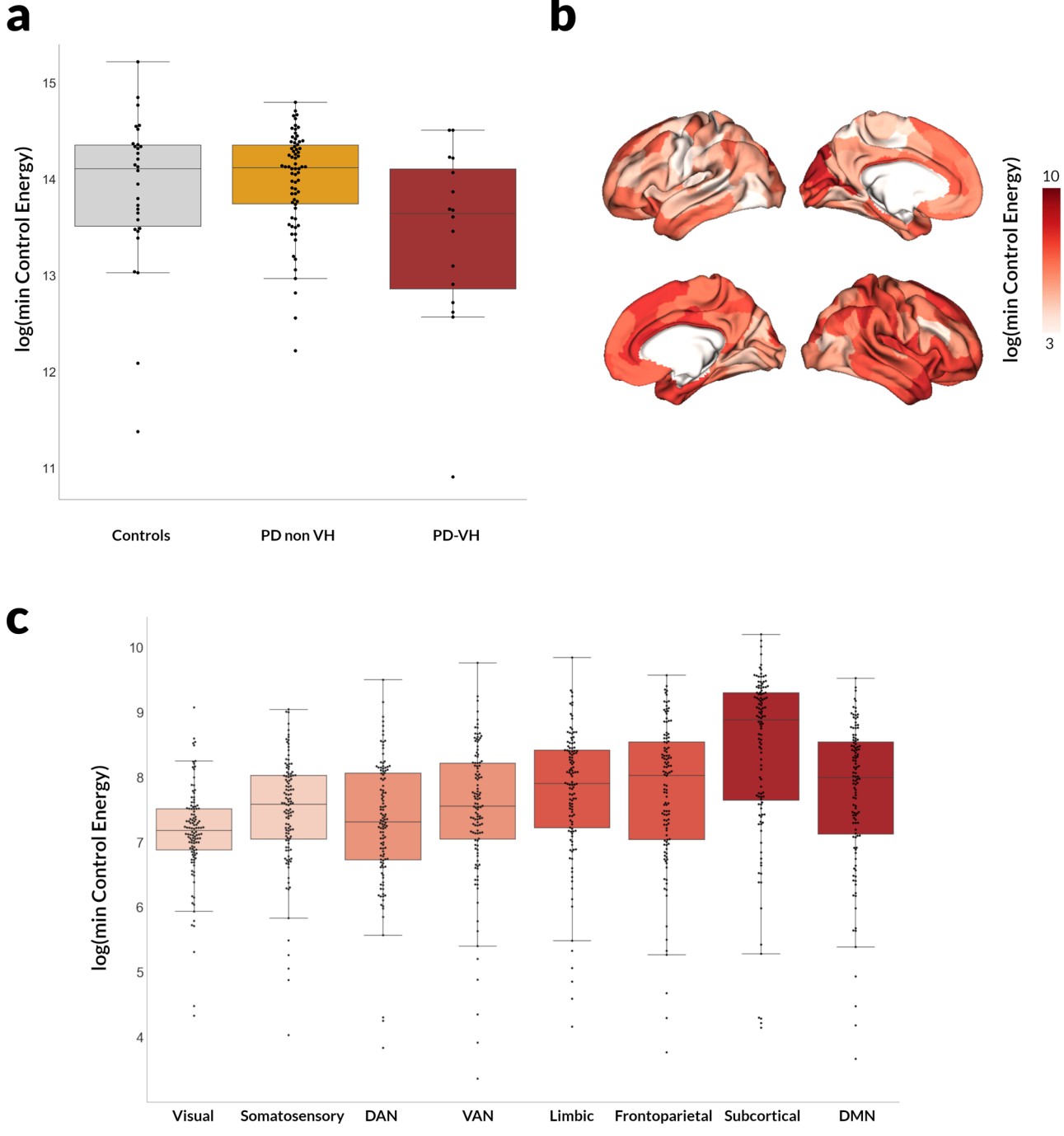

**Fig. 3 Changes in control energy to transition from the *Integrated* to the *Segregated* state in patients with Parkinson's and visual hallucinations.** **a** Minimal control energy to transition from the *Integrated* to the *Segregated* state Less energy is needed to transition for patients with Parkinson's and visual hallucinations (PD-VH, $n = 16$) than those without hallucinations (PD-non-VH, $n = 75$). Log-transformed minimal control energy is presented. Error bars are 95% confidence intervals. **b** Regional variation in minimal control energy to transition from the *Integrated* to the *Segregated* state The log-transformed minimal control energy that needs to be applied to each node ($n = 232$ nodes) is presented; darker colours denote higher amounts of energy required. Note that only cortical regions are plotted. **c** Minimal control energy per functional subnetwork. The mean minimal control energy to transition from the *Integrated* to the *Segregated* state across all nodes ($n = 232$ nodes) of the seven cortical and one subcortical resting state networks is plotted. Darker colours denote higher levels of the cortical hierarchy; also left to right: unimodal to transmodal regions. There was a significant correlation between the minimal transition energy from integrated-to-segregated state that was needed to be applied to each node and the nodes position in the cortical hierarchy, with higher amount of energy needed for more transmodal regions ($\rho = 0.526$, $p < 0.001$). Error bars are 95% confidence intervals.

energy needed to transition from *Integrated*-to-*Segregated* state in PD-VH). As expected[51], subcortical regions were strongly represented, with 25 subcortical nodes amongst the top 20% of contributors (25/47 or 53.2%) with thalamic regions amongst the highest contributors. Of the cortical nodes, top contributors included predominantly right hemispheric regions (20/22 cortical nodes) including regions of the Default mode network: cingulum, precuneus, inferior and superior temporal regions and medial frontal regions (Table 3 and Fig. 3b). There was a significant correlation between the *Integrated*-to-*Segregated* state transition

**Table 3 Top 20% of nodes that contribute to the transition from the *Integrated* to the *Segregated* state of dynamic functional connectivity.**

| Region | Coordinates in MNI space | | | Log(Energy) mean (std) | Network |
|---|---|---|---|---|---|
| | *x* | *y* | *z* | | |
| *Cortical* | | | | | |
| Occipital_Mid_L | −22 | −96 | 6 | 9.66 (8.49) | Visual |
| Cuneus_L | −12 | −72 | 22 | 9.65 (8.49) | Visual |
| Temporal_Sup_R | 64 | −24 | 8 | 9.70 (8.35) | Somatosensory |
| Temporal_Sup_R | 44 | −28 | 18 | 9.72 (8.81) | Somatosensory |
| Rolandic_Oper_R | 60 | 0 | 10 | 9.59 (8.60) | Somatosensory |
| Postcentral_R | 58 | −4 | 30 | 9.63 (8.54) | Somatosensory |
| Paracentral_Lobule_R | 6 | −22 | 68 | 9.94 (8.62) | Somatosensory |
| Cingulum_Mid_R | 10 | −36 | 46 | 9.38 (8.48) | VAN |
| Temporal_Inf_R | 46 | −12 | −34 | 9.79 (8.38) | Limbic |
| ParaHippocampal_R | 26 | −10 | −32 | 9.59 (8.38) | Limbic |
| SupraMarginal_R | 62 | −38 | 36 | 8.73 (8.76) | Frontoparietal |
| Temporal_Inf_R | 62 | −42 | −12 | 9.38 (8.60) | Frontoparietal |
| Cuneus_R | 14 | −70 | 36 | 9.39 (8.60) | Frontoparietal |
| Cingulum_Mid_R | 6 | −24 | 30 | 8.99 (8.78) | Frontoparietal |
| Cingulum_Mid_R | 4 | 2 | 30 | 7.97 (7.90) | Frontoparietal |
| Cingulum_Ant_R | 8 | 30 | 28 | 7.64 (7.77) | Frontoparietal |
| Angular_R | 50 | −58 | 44 | 9.49 (8.50) | DMN |
| Rectus_R | 4 | 36 | −14 | 9.59 (8.38) | DMN |
| Cingulum_Ant_R | 8 | 42 | 4 | 8.74 (8.94) | DMN |
| Frontal_Sup_Medial_R | 8 | 58 | 18 | 5.07 (5.50) | DMN |
| Frontal_Mid_R | 28 | 30 | 42 | 7.92 (8.18) | DMN |
| Precuneus_R | 6 | −58 | 44 | 9.24 (8.41) | DMN |
| *Subcortical* | | | | | |
| Anterior hippocampus R | 26 | −14 | −20 | 9.55 (8.57) | Subcortical |
| Posterior hippocampus R | 28 | −32 | −8 | 9.68 (8.87) | Subcortical |
| Lateral amygdala R | 28 | −2 | −22 | 9.83 (8.30) | Subcortical |
| Medial amygdala R | 22 | −6 | −16 | 5.49 (5.59) | Subcortical |
| Dorsoposterior thalamus R | 16 | −30 | 2 | 9.86 (8.64) | Subcortical |
| Ventroanterior thalamus R | 8.0 | −10.0 | 6.0 | 9.22 (8.67) | Subcortical |
| Dorsoanterior thalamus R | 12.0 | −22.0 | 12.0 | 8.57 (8.19) | Subcortical |
| Nucleus accumbens, shell R | 12.0 | 10.0 | −6.0 | 9.61 (8.68) | Subcortical |
| Nucleus accumbens, core R | 14.0 | 18.0 | −2.0 | 9.05 (8.53) | Subcortical |
| Posterior globus pallidus R | 24.0 | −8.0 | −2.0 | 7.55 (8.16) | Subcortical |
| Posterior Putamen R | 30.0 | −6.0 | 4.0 | 5.15 (5.45) | Subcortical |
| Anterior Caudate R | 14.0 | 14.0 | 6.0 | 8.05 (7.60) | Subcortical |
| Posterior Caudate R | 14.0 | 4.0 | 16.0 | 9.89 (9.13) | Subcortical |
| Anterior hippocampus L | −24.0 | −14.0 | −20.0 | 8.15 (8.26) | Subcortical |
| Posterior hippocampus L | −26.0 | −32.0 | −8.0 | 9.32 (8.23) | Subcortical |
| Lateral amygdala L | −26.0 | −2.0 | −22.0 | 8.36 (8.29) | Subcortical |
| Medial amygdala L | −20.0 | −6.0 | −16.0 | 9.67 (8.64) | Subcortical |
| Dorsoposterior thalamus L | −14.0 | −30.0 | 2.0 | 9.74 (8.39) | Subcortical |
| Ventroanterior thalamus L | −6.0 | −10.0 | 6.0 | 8.22 (8.06) | Subcortical |
| Dorsoanterior thalamus L | −10.0 | −22.0 | 12.0 | 7.47 (6.76) | Subcortical |
| Nucleus accumbens, shell L | −10.0 | 10.0 | −6.0 | 7.66 (7.69) | Subcortical |
| Nucleus accumbens, core L | −12.0 | 18.0 | −2.0 | 9.30 (8.50) | Subcortical |
| Posterior globus pallidus L | −22.0 | −8.0 | −2.0 | 7.30 (7.50) | Subcortical |
| Anterior Caudate L | −12.0 | 14.0 | 6.0 | 9.87 (8.63) | Subcortical |
| Posterior Caudate L | −12.0 | 4.0 | 16.0 | 6.89 (7.14) | Subcortical |

*L* left, *R* right hemisphere.

energy required at each node and the node's position in the cortical hierarchy, with higher energy needed for more transmodal regions ($\rho = 0.526$, $p < 0.001$) (Fig. 3c).

**Correlation with neurotransmitter systems.** Finally, we examined whether the *Integrated*-to-*Segregated* state transition (which was the state transition that specifically differed for PD-VH patients), is associated with specific neurotransmitter systems (Supplementary Table 2) in the healthy brain. To do this, we correlated the mean control per node to transition from the *Integrated*-to-*Segregated* state with mean regional neurotransmitter density (derived from open-access PET data) and neurotransmitter receptor gene expression levels (derived from the Allen Brain atlas[52]) in health; we tested this against spatially-correlated null models through sphere permutations, FDR-corrected for multiple comparisons over 232 nodes (Fig. 1c).

We found a significant correlation between regional log(Energy) and density of *5HT1b* ($\rho = -0.274$, $q_{spin} = 0.009$), *5-HT2a* ($\rho = -0.347$, $q_{spin} < 0.001$) and *GABA_A* receptors ($\rho = -0.317$, $q_{spin} = 0.022$), from open-access atlases of PET data (Fig. 4). Regional energy and regional expression levels of genes relating to 5-HT2a receptors were also significantly correlated ($\rho = -0.1438$,

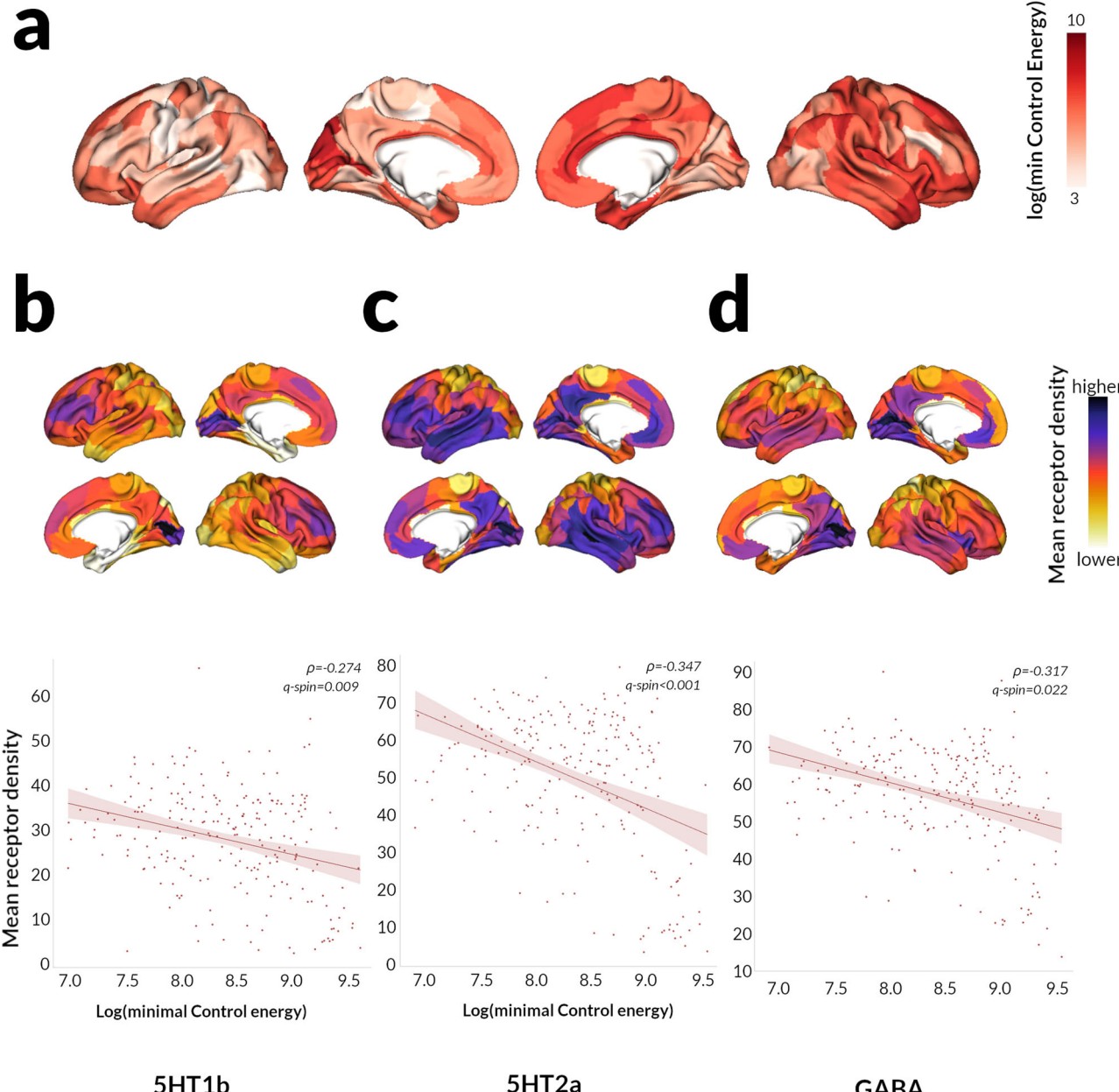

**Fig. 4 Neurotransmitter correlates of Integrated-to-Segregated state transition.** The log-transformed minimal control energy that needs to be applied to each node ($n = 232$ nodes) to achieve the Integrated-to-Segregated state transition (**a**) was correlated with the mean regional receptor density of 5HT1b receptors (**b**), 5HT2a receptors (**c**) and GABA receptors (**d**), from open access atlases of PET data in unaffected individuals. In all cases, $\rho$ is the Spearman correlation coefficient and $q$-spin is the FDR-corrected $p$-value derived following spatial permutations ($p$-spin, 1000 permutations).

$q_{spin} = 0.044$) as well as two $GABA_A$ receptors [GABRA1 ($\rho = -0.2437$, $q_{spin} = 0.020$) and *GABRA2* ($\rho = 0.128$, $q_{spin} = 0.023$)]; gene expression data for 5-HT1b receptors were not available. Although noradrenergic and acetylcholinergic PET data are not publicly available, genetic expression of noradrenergic (*ADRA1B* and *ADRA2A*), muscarinic (*CHRM1, CHRM2, CHRM3, CHRM4*) and nicotinic receptors (*CHRNA3, CHRNA4, CHRNA7, CHRNB2*) was correlated with regional transition energy. Gene expression of DRD2 was also correlated with regional control energy for the *Integrated*-to-*Segregated* state transition ($\rho = 0.318$, $q_{spin} = 0.013$) but this was not replicated using density PET-derived data ($\rho = 0.056$, $q_{spin} = 0.800$). The detailed correlations between regional control energy and transmitter density and regional gene expression are seen in Table 4.

## Discussion

We have used dynamic functional connectivity and network control theory to explore the temporal dynamics underlying visual hallucinations in Parkinson's, and examined how these can be explained through changes in brain structure. We found that PD-hallucinators spent more time in a *predominantly Segregated* state of functional connectivity than those without hallucinations, with fewer total transitions and longer dwelling time within the *Segregated* state. The transition from the *Integrated*-to-*Segregated* state was less energy-demanding in PD-hallucinators than non-hallucinators. This transition is mediated by transmodal brain regions that are associated with specific neurotransmitter systems, as confirmed through both in vivo PET mapping and post-mortem gene expression microarray data.

**Table 4 Neurotransmitter receptors showing density and gene expression correlations with regional control energy required to transition from the _Integrated_ to the _Segregated_ state.**

| Receptor | Ligand | Correlation coefficient | q value |
|---|---|---|---|
| _Receptor density_ | | | |
| 5-HT1B | Serotonin | −0.274 | 0.009 |
| 5-HT2A | Serotonin | −0.347 | 0.000 |
| GABA | GABA | −0.317 | 0.022 |

**_Receptor gene expression_**

| Gene symbol | Ligand | Correlation coefficient | q value |
|---|---|---|---|
| ADRA1B | Norepinephrine | −0.154 | 0.018 |
| ADRA2A | Norepinephrine | −0.210 | 0.013 |
| CHRM1 | Acetylcholine | −0.279 | 0.018 |
| CHRM2 | Acetylcholine | −0.265 | 0.028 |
| CHRM3 | Acetylcholine | −0.223 | 0.018 |
| CHRM4 | Acetylcholine | 0.202 | 0.018 |
| CHRNA3 | Acetylcholine | 0.416 | 0.013 |
| CHRNA4 | Acetylcholine | −0.158 | 0.033 |
| CHNRA7 | Acetylcholine | −0.244 | 0.023 |
| CHNRB2 | Acetylcholine | 0.207 | 0.028 |
| DRD2 | Dopamine | 0.318 | 0.013 |
| HTR1E | Serotonin | −0.207 | 0.013 |
| HTR1F | Serotonin | −0.3301 | 0.013 |
| HTR2A | Serotonin | −0.144 | 0.044 |
| HTR5A | Serotonin | −0.311 | <0.001 |
| GABRA1 | GABA | −0.244 | 0.020 |
| GABRA2 | GABA | 0.128 | 0.023 |
| GABRAB2 | GABA | 0.433 | 0.018 |
| GABRAD | GABA | −0.289 | 0.013 |
| GABRG1 | GABA | 0.227 | 0.023 |
| GABRG2 | GABA | −0.337 | 0.023 |
| GABRG3 | GABA | −0.217 | 0.044 |

Note that correlation coefficients of absolute values between 0.1 and 0.4 represent moderate correlation in our dataset. Q values are FDR-corrected _p_-values from spatial permutation testing (q-spin).
_CHRNA_ nicotinic cholinergic receptor (Alpha), _DRD_ dopamine receptor D, _HTR_ 5-hydroxytryptamine receptor, _ADRA_ alpha-1A adrenergic receptor, _GABR_ GABA receptor.

Previous studies have shown that PD patients with cognitive impairment similarly spend more time in a Segregated state and show fewer transitions between states than PD with intact cognition and controls[23,24]. There were no differences in cognitive performance between PD patients with and without hallucinations in our cohort, but visual hallucinations are known to be associated with incipient dementia in PD[53]. In schizophrenia, where auditory hallucinations are a core feature, similar findings of altered dwell time are seen[14,54], correlated with severity of hallucinations[55]. We similarly saw patients with visual hallucinations spending less time in the _Integrated_ (and more time in the _Segregated_) state suggesting this finding may be specific to hallucinations as a trait.

We found that only the temporal dynamics of functional connectivity were altered in patients with hallucinations. This indicates that a change in the temporal balance between normal/preserved states rather than a change in the states themselves underlie PD-hallucinations. This contrasts with work using similar methodologies in patients with loss of consciousness and in healthy volunteers after LSD administration where within-state changes particularly within the _Integrated_ state[18,20], were also seen. However, as we examined the propensity to hallucinate rather than the hallucinatory state itself (participants were not actively experiencing hallucinations during scanning) it is possible that additional within-state changes could underlie visual

hallucinations in PD, in the moment when they actually occur, an avenue for potential future investigations. In addition, although hallucinations in our participants were frequent (at least weekly in most participants) they were not universally complex and severe. Other important differences are that LSD-induced visual hallucinations are associated with changes in other sensory modalities including time/space dysperceptions and ego dissolution[56], which are not seen with PD-associated hallucinations; thus it is not unexpected that the underlying changes in temporal dynamics are different between these hallucinatory conditions.

As temporal transition between functional states is constrained by structural connectivity[31,33,57], we used network control theory to model the ease and regional contribution to the brain's activation for each of these two states, represented as the vector of sum connectivity for each region at each time point[31]. Specifically, we examined the energy cost of transitioning between and maintaining the _Integrated_ and _Segregated_ states. Minimisation of the control energy cost to transition into a state will make this transition more preferable, and evidence in healthy adolescents suggests lower control energy to activate the frontoparietal network during development (secondary to structural connectome reconfiguration) supports improved executive function[35]. In addition, hallucination-inducing substances such as LSD and psilocybin have been recently shown to reduce overall control energy needed for between brain-state transitions leading to a more temporal labile functional landscape[58]. We found a significantly lower energy cost to transition from the _Integrated_-to-_Segregated_ state for PD-hallucinators than non-hallucinators. In this way, network control theory provides mechanistic insights about why patients with PD-VH spend more time in a more Segregated state: as it is more energy efficient to transition from the Integrated-to-Segregated state due to constraints caused by loss of structural integrity. Further, this framework enabled us to identify the particular nodes most critical in mediating these transitions, with subcortical (especially thalamic nodes) and regions within the DMN especially implicated, consistent with previous work highlighting DMN involvement in PD hallucinations[6]. Thalamic regions were amongst the highest contributors to this transition. Thalamic involvement has been previously described in visual hallucinations[50,59] and we recently showed longitudinal changes in grey and white matter within the medial mediodorsal thalamus[60]. This provides further evidence of the thalamus as a key driver of network imbalance in PD-hallucinations[51,61].

Interestingly, the brain regions contributing most to this transition from _Integrated_-to-_Segregated_ state showed a correlation with specific neurotransmitter systems in health. Although the directionality of the relationship is difficult to interpret as data on regional neurotransmitter density and gene expression were derived from healthy individuals, regional density of _5HT2A_ receptors was significantly correlated with the regional control energy needed for _Integrated_-to-_Segregated_ state transition; this was replicated using regional expression data for the _5HT2A_ receptor gene.

Activation of 5HT2A receptors is a key mechanism for drug-induced hallucinations occurring with the psychedelic drugs, LSD, psilocybin and ayahuasca[62] and modelling studies have shown that this receptor plays a key role in engendering the characteristic brain dynamics of LSD[63]. Recent work highlighted the crucial role of 5HT2A in neuronal-neurotransmission dynamic coupling across the brain[37]. 5HT2A has also been implicated in PD-hallucinations; evidenced by the higher density of 5HT2A receptors within frontal, temporal and occipital regions in patients with PD hallucinations in post mortem and in vivo studies[43,64] and the efficacy of the novel 5HT2A inverse agonist

Pimavanserin in the treatment of PD-hallucinations[65]. Visual hallucinations are a common phenomenological endpoint of both LSD and PD; our findings provide further evidence for the role of 5HT2A involved in PD-hallucinations, suggesting a convergent biological substrate across hallucinations irrespective of cause.

Other serotonergic receptors were also important for the *Integrated-to-Segregated* state transition including: 5HT1B (receptor density, but no genetic expression data), 5HT1E, 5HT1F and 5HT5A (gene expression data only). The correlation with multiple serotonin receptors, indicates that serotonergic modulators targeting multiple receptors could be potential therapeutic targets for PD-hallucinations. Of note, no receptor density or gene expression data were available for 5HT3 receptors, a target of interest for Ondansetron, a 5HT3-antagonist currently under evaluation as a treatment of hallucinations[66].

Regional receptor density and gene expression for GABAergic receptors were also correlated with regional transition energy in line with previous studies showing reduced GABA concentration in the visual cortex of PD-hallucinators[44,67]. Visual processing involves a complex interplay between monoaminergic, cholinergic and GABA/glutamatergic neurotransmission[61]. The observed correlation between the *Integrated-to-Segregated* state transition and regional gene expression of noradrenergic (ADRA1B, ADRA2A) and cholinergic (muscarinic and nicotinic) receptors is consistent with this, but there were no available PET-derived density data to replicate this.

Convergent evidence has recently highlighted the importance of the noradrenergic system in some non-motor PD symptoms[68–70]. Noradrenaline plays a key role in modulating selective attention[71] and with serotonin, modulates behavioural responses to incoming visual information[61]. The noradrenergic system is also likely to play a key role in mediating functional state transitions: noradrenaline-mediated apical amplification of pyramidal cells differentiates waking and anaesthesia[72], extracellular noradrenaline is associated with sleep-state transitions[73] and locus coeruleus activity flexibly mediates the recruitment of other neural circuits particularly the prefrontal cortex[74], leading to dynamic changes in functional networks, specifically transitioning between motor and task-negative networks[75]. Changes within the noradrenergic system may be involved in altered state transitions in PD-hallucinations by modulating the activity of sensory cortices and thalamocortical neurocircuitry[76].

In contrast to these other neurotransmitters, we found no consistent correlation with dopaminergic receptors. It is important to note that although DRD1 is one of the major dopamine receptors in the cortex no DRD1 density data was publicly available at the time of the study; however, no correlation was seen between genetic expression of DRD1 and regional contribution to the *Integrated-to-Segregated* state transition. Although a lack of correlation between regional dopamine expression and regional energy does not exclude an indirect dopamine effect in visual hallucinations, our findings highlight the role of transmitters other than dopamine in the development of PD-hallucinations. Rather than a simple hyperdopaminergic state leading to PD-hallucinations, our findings suggest a complex imbalance in multiple neurotransmitter systems, with changes in 5HT2A, GABA and noradrenergic receptors all contributing. Treatment options targeting more than one neurotransmitter system may therefore be needed to manage visual hallucinations in PD and other psychotic illnesses.

Several considerations need to be taken into account when interpreting our findings. Our sample size of visual hallucinators is small, although comparable with other published studies[24,27]. Functional data are susceptible to motion artefact; we adopted strict exclusion criteria to mitigate for this[77] and motion as well as image quality metrics did not differ between groups. We chose not to perform global signal regression in keeping with other studies using the same analyses[18,20]. Although this can be used to counteract residual artefacts from head motion[77] it can contain behaviourally-relevant information and affect group results[78,79], and we instead adopted stringent exclusion criteria for motion to limit potential motion effect. All participants were scanned while receiving their usual dopaminergic medications and at the same time of day and levodopa equivalent doses did not significantly differ between PD-VH and PD-non-VH[80]. Further studies assessing PD patients ON and OFF levodopa might provide additional information. Although brain networks are non-linear, we used a linear optimal control model since this has been shown to provide important insights into non-linear dynamics[81] and linear-Gaussian models are often adequate descriptors of functional MRI timeseries, such that more complex, non-linear models often do not provide additional explanatory power[82,83]. Nevertheless, future work may seek to leverage insights from non-linear models of brain dynamics, e.g. through neurobiologically detailed *dynamic mean-field* models that have already been successfully applied to the study of altered states of consciousness[63,84]. Most studies using network control theory so far have assessed transitions from rest to task[34] or activation of a specific functional network[35] where brain states are defined as regional activation by selecting which regions should be active or inactive. However, as our key question was to examine the transition between the integrated and segregated states (identified from the dynamic changes in the resting state timeseries from our patients), of dynamic functional connectivity in our patients, such an approach was not straightforwardly applicable in our case. Instead, in the literature on brain-states, it is common to define data-driven brain states in terms of time-resolved patterns of functional connectivity: that is, states are defined in terms of how regions are co-active together over a short period of time, rather than by specifying which regions should be active. Therefore, we used the sum of regional functional connectivity as a summary representation for each previously identified state; in this setting our activity states correspond to brief periods where specific nodes are active or inactive together, rather than nodes having high or low activity per se. In other words, our approach identifies states in terms of nodal co-activation. Although this approach provides insights into the relative ease of each transition, a task vs rest approach would be potentially even more informative and could be examined in future work. Finally, data on neurotransmitter density and gene expression were not derived from our participants but from separate cohorts of healthy volunteers and post-mortem human brains, respectively; therefore results relating to neurotransmitter receptors should be interpreted with caution. Future work may seek to replicate these results with each patient's own unique neurotransmitter receptor signature, which may offer individualised insights and the opportunity to assess the directionality of this relationship, as well as potential targets for pharmacological intervention.

Our findings describe that temporal functional dynamics are altered in PD-hallucinations, with a predisposition towards a *Segregated* state of functional connectivity. This segregated state predominance can be explained by a reduced energy cost to transition from the integrated-to-segregated state in PD patients with hallucinations compared to those without hallucinations. We have also clarified the neuromodulatory correlates of the integrated-to-segregated state transition in the healthy brain. These results provide mechanistic insights into visual hallucinations in PD with implications for other psychotic disorders. By linking these changes to neurotransmitter systems, our findings highlight possible therapeutic targets for hallucinations, a core symptom of psychosis.

## Methods

**Participants.** 123 participants were included: 91 PD patients and 32 unaffected controls. The study was approved by the Queen's Square Ethics Committee and participants provided written informed consent. Patients with PD were classified as PD with visual hallucinations (PD-VH, $n = 16$) if they scored ≥1 in Question 2.1 of the Unified Parkinson's Disease Rating Scale (UPDRS); the rest were classified as PD-non-VH ($n = 75$). We collected additional information on severity, frequency and phenomenology of experienced hallucinations with the University of Miami Parkinson's Disease Hallucinations Questionnaire (UM-PDHQ)[85]. General cognition was assessed using the Mini-Mental State Examination (MMSE) and Montreal Cognitive Assessment (MoCA)[86,87]. In addition, domain-specific cognitive assessments with two tests per domain included: Attention: Digit span backwards[88], Stroop, Naming[89]; Executive functions: Stroop Interference[89], Category fluency[90]; Memory: Word Recognition Task[91], Logical Memory[88]; Language: Graded Naming Task[92], Letter fluency[90]; and Visuospatial: Benton's Judgement of Line[93], Hooper Visual Organization Test[93]. Mood was assessed using the Hospital Anxiety and Depression Scale (HADS)[94]. Disease-specific measures for PD included: motor assessment using the Movement Disorder Society UPDRS[95], smell: Sniffin' Sticks[96], and sleep: REM Sleep Behaviour Disorder Questionnaire (RBDSQ)[97]. Levodopa equivalent daily doses (LEDD) were calculated for PD participants[98].

**MRI data acquisition and preprocessing.** Imaging data were acquired on the same 3T Siemens Prisma-fit scanner: rsfMRI: gradient-echo EPI, TR = 70 ms, TE = 30 ms, 105 volumes; diffusion-weighted (DWI): 64 directions (b-values: 50, 300, 1000, 2000). Scanning took place at the same time of day, with PD patients receiving their normal anti-Parkinsonian medication.

Both imaging modalities underwent rigorous quality assurance: The MRI quality control tool (MRIQC) was used to assess rsfMRI data[99]. Participants were excluded if any of the following was met: (1) mean frame-wise displacement >0.3 mm, (2) any frame-wise displacement >5 mm, or (3) outliers >30% of the whole sample. This led to 12 participants being excluded (11 PD). Therefore, 91 patients with PD (16 PD-VH and 75 PD-non-VH) and 32 controls are included. Note that our sample includes patients that overlap with other reports from our centre. Slight differences in included patients are caused by exclusions such as head movement and quality control that differ between studies.

All of raw DWI datasets were visually inspected and evaluated for artefact; only scans with <15 volumes containing artefacts[100] were included in subsequent structural analyses, resulting in further 5 PD and 2 control participants being excluded.

Preprocessing of rsfMRI data was performed as described previously[101]. In brief, we used fMRIPrep 1.5.0[102] and discarded the first 4 volumes to allow steady-state equilibrium. Functional data were slice-time corrected using 3dTshift from AFNI[103] and motion corrected using mcflirt[104]. Distortion correction was performed using TOPUP[105]. This was followed by co-registration to the corresponding T1-weighted image using boundary-based registration with six degrees of freedom[106]. Motion correcting transformations, field distortion correcting warp, BOLD-to-T1w transformation and T1w-to-template (MNI) warp were concatenated and applied in a single step using antsApplyTransforms (ANTs v2.1.0) using Lanczos interpolation. Physiological noise regressors were extracted applying CompCor[107]. Spurious sources of signal were removed through linear regression: six motion parameters, mean signal from white matter and cerebrospinal fluid. We did not regress global signal given the lack of consensus and potential to distort group differences[78].

Preprocessing of diffusion-weighted images was performed in MRtrix3.0[108] using dwipreproc, with denoising[109], removal of Gibbs ringing artefacts[110], eddy-current and motion correction[111] and bias field correction[112].

**Parcellation.** To construct functional and structural connectivity matrices, each participant's T1-weighted image was parcellated into 200 cortical and 32 subcortical regions of interest (ROIs) using the Schaefer[113] and Tian parcellations[114], respectively. Parcellations of 200 regions result in connectomes with the highest representativeness[115,116] and the combined Schaefer-232 parcellation used here, is considered optimal across structural and functional connectomes[115]. We used the same parcellation to construct functional and structural connectivity matrices for each participant. To ensure robustness of results, analyses were replicated using the finer-grained Schaefer/Tian parcellations with 400 cortical and 54 subcortical ROIs, respectively.

**Dynamic functional connectivity analysis.** Dynamic connectivity matrices were derived using an overlapping sliding-window approach[10] with windows of 44 s duration (63*TR, within the recommended range[10]) in steps of 1 repetition size (63 windows 44 s each) (Fig. 1a). A 232*232 weighted adjacency matrix representing the functional connectome for that time point was calculated for each window.

We then identified states of higher integration or segregation using a "cartographic profile"[12,18,20,117]. At each time point, the asymmetric algorithm of Rubinov and Sporns[118] was used to identify network modules by applying the community Louvain algorithm, which iteratively evaluates different ways of assigning nodes to modules, in order to maximise the resulting modularity function Q:

$$Q = \frac{1}{v^+} \sum_{ij} \left( w_{ij}^+ - e_{ij}^+ \right) \delta_{M_i M_j} - \frac{1}{v^+ + v^-} \left( w_{ij}^- - e_{ij}^- \right) \delta_{M_i M_j} \quad (1)$$

where $v$ is the total weight of the graph (sum of all the graph's edges), $w_{ij}$ is the signed weight of the edge between nodes $i$ and $j$, $e_{ij}$ is the weight of an edge divided by the total weight of the graph (superscripts denote + positive and − negative edges), and $\delta_{M_i M_j}$ is set to 1 when nodes $i$ and $j$ are in the same module and 0 otherwise. We performed 100 iterations for each time-resolved network with module size resolution parameter $\gamma$ set at the default $\gamma = 1$.

We calculated participation coefficient and within-degree Z-score for each node using the Brain Connectivity Toolbox. Participation coefficient was calculated as:

$$P_i = 1 - \sum_{s=1}^{M} \left( \frac{\kappa_{is}}{k_i} \right)^2 \quad (2)$$

where $\kappa_{is}$ is the strength of positive connections between node $i$ and other nodes in module $s$; $k_i$ is the strength of all its positive connections; and $M$ is the number of modules in the network, as identified by the modularity detection algorithm. The participation coefficient ranges between zero (no connections with other modules) and one (equal connections to all other modules). High mean participation coefficient within a network implies higher levels of integration between-modules.

The within-module degree Z-score $Z_i$ was calculated as:

$$z_i = \frac{\kappa_{is} - \bar{\kappa}_{is}}{\sigma_{\kappa_{is}}} \quad (3)$$

where $\kappa_{is}$ is the strength of connections between node $i$ and other nodes in module $s$, and $\bar{\kappa}_{is}$ and $\sigma_{\kappa is}$ are, respectively, the average and the standard deviation of $\kappa_{is}$ over all nodes belonging to module $s$.

Joint histograms of participation coefficient and within-module Z-score were then derived for each time point[12] and for each participant. The cluster with the higher average participation coefficient was defined as the "Integrated" state and the cluster with the lower average participation coefficient as the "Segregated" state, as previously described[12,18,20]. K-means clustering was then performed and assigned each dynamic functional connectivity matrix to one of two clusters (Integrated vs Segregated)[12,18,20] (Fig. 1a). K = 2 clusters was also best performing in data-driven evaluation (Supplementary Fig. 1).

We calculated: (1) proportion of time spent in each state as the number of timepoints within each state divided by number of total timepoints(63), (2) average dwell time as the number of consecutive windows/timepoints belonging to each state and (3) number of transitions as the number of transitions from one state to the other; transitions were further divided into transitions from integrated-to-segregated and from segregated-to-integrated states.

**Structural network construction.** After DWI-image preprocessing, diffusion tensor metrics were calculated for each participant and constrained spherical de-convolution performed[119] followed by anatomically constrained tractography (10 million streamlines)[120] and spherical de-convolution-informed filtering of tracto-grams (SIFT2)[121]. The resulting set of streamlines, weighted by a cross-sectional multiplier, was used to construct the structural brain network as a 232*232 undirected weighted connectivity matrix.

**Network control analysis.** We examined how the structural brain network of each participant, composed of white matter tracts, constrains the brain in transitioning from one state of functional connectivity (Integrated or Segregated) to the other. To do this, we used a linear time-invariance network model, as previously detailed[29,31,50]. This can describe neural states as simulated states ($x$) of a network with $n$ nodes over time steps $t$ using:

$$x(t+1) = Ax(t) + Bu(t) \quad (4)$$

where $x(t)$ is a vector (1*$n$ nodes) that represents the brain state at given time $t$, $n$ is the number of nodes (232 ROIs), matrix $A$ represents the structural connectome $n*n$ (normalised to ensure stability[31,32]), matrix $B$ is the matrix of control nodes for the network with $n*n$ dimensions and $u(t)$ is the control energy applied for each node at a given time $t$. In all analyses, we did not constrain the number of nodes that could be controlled, therefore B is an identity matrix.

This model can be used to derive the structural control energy necessary to transition from an initial state $x(0)$ to a target state $x(T)$ where $T = 1$ is the control horizon[31,122] as:

$$\min_u \int_0^T (x_T - x(t))' S(x_T - x(t)) + \rho u(t)' t(t) dt \quad (5)$$

where $x_T$ is the target state (1*$n$ vector where $n$ is the number of nodes), $S$ is the diagonal $n*n$ matrix that selects a subset of states to constrain (here the identity matrix), $\rho$ is the importance of the input penalty to the state penalty (here $\rho = 1$) and $T$ is the control horizon.

Importantly, this formalism does not prescribe how the initial "brain state" $x(0)$ and target state $x(T)$ are identified: both data-driven and pre-specified states have been used[33,35]. Here, our goal is to provide a mechanistic understanding of dynamic transitions between the Integrated and Segregated states of dynamic

functional connectivity. Since the network control theory model requires each state to be represented as a 1*232 vector, we represented the *Integrated* and *Segregated* states by their sum connectivity profiles (or "connectivity density"), comprised of the sum of the connection weights (Pearson correlation coefficient) from each node to all other nodes; this was calculated separately for each state.

We then used this equation to calculate the control energy to be applied to each node of the network to: (1) transition from the integrated-to-segregated state (using $x_0$ (baseline state), the sum connectivity vector of the *Integrated* state; and $x_T$ (target state), the sum connectivity vector of the *Segregated* state), (2) transition from the segregated-to-integrated state, using as $x_0$ the sum connectivity vector of the *Segregated* state and state $x_T$, the sum connectivity vector of the *Integrated* state, and (3) persist within the *Integrated* or within the *Segregated* state (i.e. transition from one state to itself), using the sum connectivity vector for that state for both $x_0$ and $x_T$ (Fig. 1b). A sum of the control energies to be applied across all nodes of the network represents the minimal energy for the specific transition. Thereby, minimal transition and persistence energies were calculated for each individual's own *Integrated* and *Segregated* functional state by capitalising on the availability of both functional and structural data for each individual.

We also identified which brain regions contribute more to the transition from the *Integrated*-to-*Segregated* state (which differed between PD-VH and PD-non-VH), or which nodes require more energy to be applied to them in order to transition: high contributors to the state transition were defined as the top 20% of regions.

**Statistics and reproducibility**. Between-group differences in clinical characteristics and temporal properties of dynamic states were assessed using ANOVA (post hoc Tukey) or *t*-tests for normally distributed and Kruskal–Wallis (post hoc Dunn) or Mann–Whitney for non-normally distributed variables (normality assessed using Shapiro-Wilk test and visual inspection). Statistical significance threshold $p < 0.05$. Differences in transition and persistence energy between PD-VH vs PD-non-VH were performed using repeated measures ANOVA ($p < 0.05$).

In addition, we investigated whether each of the two states significantly differed across groups using network-based statistics (NBS)[123]. A general linear model was used with PD-VH versus PD-non-VH and PD versus controls as contrasts of interest and age and total intracranial volume as covariates. Permutation testing with unpaired *t*-tests was performed (5000 permutations), calculating a test statistic for each connection. An a priori threshold of $t = 2.7$ was applied based on our sample size and family-wise error rate (FWE) of $p < 0.05$.

*Correlation with Neurotransmitter systems*. We investigated whether temporal changes in functional connectivity were associated with specific neurotransmitter systems (Fig. 1c). First, we calculated the regional control energy needed to transition towards and persist within a state that was more predominant in PD-VH. This was expressed as a vector 1*232 with one control energy value per node. Neurotransmitter profiles were extracted for each of the 232 ROIs from publically-available maps using JuSpace[124]:

- Serotonin receptors 5-HT1A, 5-HT1B, 5-HT2A based on carbonyl-(11C] WAY-100635, [(11C)P943, [(18)F]altanseri templates[125].
- D1 receptors based on the D1R-selective [11C]SCH23390 template[126].
- D2/3 receptors based on the [(11C)]raclopride template[127].
- and GABAa receptors based on the (11C)flumazenil template[128].

Each of the templates was registered to MNI space and parcellated with the Schaefer-232 atlas and mean values of binding potential were extracted from each ROI using the built-in JuSpace function[124].

Expression profiles for genes of noradrenergic, cholinergic (nicotinic and muscarinic), dopaminergic and serotoninergic receptors were obtained using data from the Allen Human Brain Atlas (AHBA)[52], with preprocessing as recently described[129]. We extracted and mapped gene expression data to the 232 ROIs of our parcellation using abagen[130]. Data was pooled between homologous cortical regions to ensure adequate coverage of both left (data from six donors) and right hemisphere (data from two donors). Distances between samples were evaluated on the cortical surface with a 2 mm distance threshold. Probe-to-gene annotations were updated in Re-Annotator[131]. Only probes where expression measures were above a background threshold in more than 50% of samples were selected. A representative probe for a gene was selected based on highest intensity. Gene expression data were normalised across the cortex using scaled, outlier-robust sigmoid normalisation. 15,745 genes survived these preprocessing and quality assurance steps. Expression profiles for 31 pre-selected genes (Supplementary Table 2) encoding receptors for noradrenaline, acetylcholine, dopamine and serotonin were extracted for each of the 232 ROIs.

We correlated regional control energy with (1) regional receptor density profiles for serotonin (*5HT1a*, *5HT2a* and *5HT1b*), dopamine (*D1* and *D2*) and *GABA* receptors, and (2) regional gene expression for 31 pre-selected genes. The significance of correspondence between regional control energy and regional neurotransmitter density/gene expression was estimated using a spatial permutation test which generates randomly rotated brain maps whilst preserving spatial covariance[132]. We performed 1000 random spatial permutations[133] and calculated the Spearman correlation coefficient between extracted regional control energy values and neurotransmitter maps to build a null distribution. The permutation-based *p*-value ($p_{spin}$) was calculated as the proportion of times that the

null correlation coefficients were greater than the empirical coefficients[132,133]. Derived $p_{spin}$ values were then corrected for multiple comparisons (Benjamini Hochberg; FDR-corrected values denoted as $q_{spin}$).

Statistical analyses were performed in Python 3 (Jupyter Lab v1.2.6).

**Reporting summary**. Further information on research design is available in the Nature Research Reporting Summary linked to this article.

## Data availability
Source data used to generate Figs. 2–4 are provided in Supplementary Data.

## Code availability
Analysis code is available here: https://github.com/AngelikaZa/TVFC. Links to further data sources and packages used are found in the Supplementary Material.

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

## Acknowledgements

We thank all our participants for their time. We gratefully acknowledge the support of NVIDIA Corporation with the donation of the Quadro P6000 GPU used for this research. The authors acknowledge the use of the UCL Myriad High Performance Computing Facility (Myriad@UCL), and associated support services, in the completion of this work. This research was also supported by the National Institute for Health Research University College London Hospitals Biomedical Research Centre. A.Z. is supported by an Alzheimer's Research UK Clinical Research Fellowship (2018B-001). P.M.C. is supported by the National Institute for Health Research. R.S.W. is supported by a Wellcome Clinical Research Career Development Fellowship (205167/Z/16/Z).

## Author contributions

Study design and concept: A.Z. and R.W., data collection: A.Z., L.A.L. and R.W., imaging and statistical analysis: A.Z., drafting and revision of the manuscript: A.Z., P.M.C., L.A.L., A.L., S.R., E.S., A.J.L. and R.W.

## Competing interests

R.S.W. has received honoraria from GE Healthcare and Britannia. The other authors declare no competing interests.
