## [Peer Review File · Communications Biology]

This manuscript has been previously reviewed at another Nature Portfolio journal. This document only contains reviewer comments and rebuttal letters for versions considered at Communications Biology.

Reviewers' comments:

Reviewer #1 (Remarks to the Author):

The authors adequately addressed my concerns and I recommend the article to be accepted.

Reviewer #2 (Remarks to the Author):

I have reviewed the manuscript before and still think it investigates a very interesting topic with innovative tools.

However, I find the three main issues raised previously have not been addressed well in the current revision:

1) Sample Size: Both reviewers have commented on the small sample size. The authors have not performed any additional analyses that can support one of their central claims, namely that changes in dynamic transition underlie visual hallucinations. The authors either need to increase their sample size or add data from other patient groups that share the transdiagnostic concept of visual hallucinations.

2) Segregated vs. integrated states: While the authors have provided additional analyses that show some differences between both states in terms of graph metrics (with clustering coefficient as an indicator of integration showing no significant difference between states), I am still missing a statistical comparison that shows that these states are actually different in terms of integration or segregation measures. Suppose the authors have just picked the cluster with the highest average participation coefficient and labeled it "integrated" without formally testing for actual statistical significance in that metric. In that case, it is highly misleading to label them as such. Integration and segregation are well-established concepts that can be formally quantified, and researchers have certain expectations if they read these terms. From the data presented so far, it looks like the clusters do differ more regarding other metrics (for example, high and low density). This should be reflected in labeling and the interpretation and discussion of results.

3) Interpretation of brain states:

Please let me clarify my questions: If you define your brain states as the sum of edges for each node, then your state vector theoretically has a unit (either strength or number of connections, etc). If you now apply control theory, the control you need to exert has the same unit, which is again connectivity strength/number of connections.

How is this conceptualization of control energy different from the usual concept of using some form of brain activity as brain states.

Response to Reviewers:

COMMSBIO-22-0549-T Changes in dynamic transitions between integrated and segregated states underlie visual hallucinations in Parkinson's disease

We thank the editors for considering our manuscript for publication in Nature Communications Biology. We were pleased that Reviewer 1 felt their comments were addressed in our revision. We now address Reviewer 2 comments 2 and 3 as per the editor's request.

In particular, we have further highlighted the a-priori definition of our predominantly "Integrated" and predominantly "Segregated" states throughout the manuscript. We also include a statistical comparison of modularity, global efficiency and participation coefficient between states in the Supplementary material.

The full details of the changes are described below.

We feel that the amendments made in response to comments from the Reviewers have improved the manuscript and hope that our revisions meet with the Editor's and Reviewers' approval.

Response to Reviewer 2:

Comment 1: *"Sample Size: Both reviewers have commented on the small sample size. The authors have not performed any additional analyses that can support one of their central claims, namely that changes in dynamic transition underlie visual hallucinations. The authors either need to increase their sample size or add data from other patient groups that share the transdiagnostic concept of visual hallucinations."*

Response: As advised by the Editor, we have not provided a response to this comment, but only to comments 2 and 3.

Comment 2: *"Segregated vs. integrated states: While the authors have provided additional analyses that show some differences between both states in terms of graph metrics (with clustering coefficient as an indicator of integration showing no significant difference between states), I am still missing a statistical comparison that shows that these states are actually different in terms of integration or segregation measures. Suppose the authors have just picked the cluster with the highest average participation coefficient and labeled it "integrated" without formally testing for actual statistical significance in that metric. In that case, it is highly misleading to label them as such. Integration and segregation are well-established concepts that can be formally quantified, and researchers have certain expectations if they read these terms. From the data presented so far, it looks like the clusters do differ more regarding other metrics (for example, high and low density). This should be reflected in labeling and the interpretation and discussion of results."*

Response: As described in the Methods, the cluster with the highest average participation coefficient was assigned as the "Integrated" and the lowest as "Segregated" state, using the cartographic profile method described by (Shine et al., 2016) and subsequently by other studies using similar methodology (Luppi et al., 2021, 2019; Shine et al., 2019; Wei et al., 2022; Zuberer et al., 2021).

The Integrated state had indeed lower characteristic path length and clustering coefficient than the segregated state, although this was not significant when correcting for multiple comparisons. We did not include initially participation coefficient in this analysis as this was chosen a-priori but we have now tested this as per the Reviewer’s request: Integrated states had on average higher participation coefficient mean (std) = 0.495 (0.128) compared to the Segregated states 0.456 (0.116), p-value: 0.013. We now include this in *Supplementary Figure 2 and Supplementary Table 3*.

Clustering coefficient and characteristic path length are amongst the most commonly used metrics quantifying segregation and integration accordingly (Rubinov and Sporns, 2010); however following the reviewers comments we have also included global efficiency as a metric of integration and modularity as a metric of segregation and clarify which are metrics of segregation and which of integration in *Supplementary Table 3 (extract below)*:

	Integrated state	Segregated state	p-value	q-value
Measures of Integration				
Characteristic path length	2.673 (2.584)	3.003 (2.284)	0.082	0.115
Global efficiency	0.958 (0.069)	0.939 (0.081)	0.043	0.077
Participation coefficient	0.495 (0.128)	0.456 (0.117)	0.013	0.045
Measures of segregation				
Clustering coefficient	0.671 (0.225)	0.692 (0.243)	0.445	0.519
Modularity	0.467 (0.118)	0.490 (0.127)	0.647	0.647
Other measures				
Small world propensity	0.427 (0.082)	0.104 (9.664)	0.002	0.014
Density	1.824 (0.277)	1.752 (0.319)	0.044	0.077
Results are mean (standard deviation) and p-value of Kruskal Wallis test. q-value: FDR corrected p-value.				

In addition, we have further pointed the a-priori definitions of states in our Methods section and, taking into account the Reviewer’s comment, we have further revised the manuscript to highlight that these states are predominantly “Integrated” and predominantly “Segregated” and chosen a-priori rather than through statistical comparison.

Comment 3: *“Interpretation of brain states: Please let me clarify my questions: If you define your brain states as the sum of edges for each node, then your state vector theoretically has a unit (either strength or number of connections, etc). If you now apply control theory, the control you need to exert has the same unit, which is again connectivity strength/number of connections. How is this conceptualization of control energy different from the usual concept of using some form of brain activity as brain states.”*

Response: Network control theory enables the quantification of the energy required to transition between states of functional brain activation. Using network control theory provides additional unique information to just using brain activity as states. Firstly, it takes into account each individual’s structural connectome, thus providing a link between structural connectivity and the observed changes in dynamic functional connectivity that we, and others, have shown in patient’s with PD and visual hallucinations (Hall et al., 2019; Zarkali et al., 2020)..

Secondly, the Integrated and Segregated states did not themselves differ between groups, only the temporal dynamics between them differed. Leveraging network control theory we can explain, using structural data this temporal difference in functional dynamics. Specifically, our findings of reduced control energy to transition from Integrated-to-Segregated state provides a mechanistic explanation for why patients with PD-VH spend more time in a segregated state (it is more energy efficient to transition from the integrated to the segregated state due to constraints caused by loss of structural integrity).

Lastly, recent work has shown that the resting human brain exhibits spontaneous tendencies towards specific state transitions over others, yet this can be overcome by cognitive demands and correlates to brain development with age and cognitive performance (Braun et al., 2021; Cornblath et al., 2020; Cui et al., 2020; Parkes et al., 2018). In addition, specifically in the study of hallucinations, a recent study showed that LSD and psilocybin both reduce the control energy required for between-state transitions leading to a more flexible temporal brain (Singleton et al., 2021). This implies that network control theory provides neurobiologically and cognitively relevant insights into brain dynamics including in the study of hallucinations.

We have now expanded our Introduction (Page 4) and Discussion sections (Pages 10-11) to further clarify the rationale for using network control theory.

Reviewers' comments:

Reviewer #2 (Remarks to the Author):

I am sorry to say that the authors have not answered my question about their brain state definition. The problem is that they define brain states in a novel and unusual way (they use a summary measure of connectivity instead of (a proxy of) activity as the previous publication have) but still discuss and interpret their findings as if they have used a measure of activity. This poses a problem in the relatively young field, because it summarizes several different approaches and potential different mechanism under one umbrella term without proper differentiation.

I have discussed this issue (anonymously and in a general way, of course) with several colleagues who have worked with control theory themselves. They all agree that this approach needs explicit mentioning and a better and explicit discussion about what this approach means for the interpretation of control energy.

Response to Reviewers:

COMMSBIO-22-0549-T Changes in dynamic transitions between integrated and segregated states underlie visual hallucinations in Parkinson's disease

We thank the Editors for considering our manuscript for publication in Nature Communications Biology and for giving us the opportunity to respond to the Reviewer's further comment.

As requested by the Editor, we have now provided a clear definition of network control theory; and included an explicit discussion for what it means for the interpretation of control theory.

We have also provided a response to the point raised by the reviewer and made further changes to the manuscript to address this. In particular, as requested, we now explicitly state in the Revised manuscript how our application of network control theory to brain states expressed as a summary measure differs from previous applications.

The full details of the changes are described below.

Response to Reviewer 2:

Comment 1: *"I am sorry to say that the authors have not answered my question about their brain state definition. The problem is that they define brain states in a novel and unusual way (they use a summary measure of connectivity instead of (a proxy of) activity as the previous publication have) but still discuss and interpret their findings as if they have used a measure of activity. This poses a problem in the relatively young field, because it summarizes several different approaches and potential different mechanism under one umbrella term without proper differentiation.*

I have discussed this issue (anonymously and in a general way, of course) with several colleagues who have worked with control theory themselves. They all agree that this approach needs explicit mentioning and a better and explicit discussion about what this approach means for the interpretation of control energy."

Response: The brain states have been defined from the functional connectivity data using a sliding window, to identify the more Integrated and Segregated state within each individual. This approach is not a novel one per se, and is now a standard approach to examine dynamic changes in functional data, with many published studies using this technique. Recent examples include Diez Cirarda 2017 and Fiorenzato, Brain 2019 applied to Parkinson's disease with dementia and mild cognitive impairment; and Luppi 2021 applied to visual hallucinations induced by LSD.

As we describe in the introduction (Page 3), the benefits of this method over previous functional connectivity measures, is that it enables us to examine changes in connections over the course of the scanning session, instead of averaging functional connectivity across the whole session. In this way, we were able to examine the dynamic changes in relative integration and segregation in patients with and without visual hallucinations, which was the primary aim of our study.

However, we agree with the reviewer that there is some novelty in applying network control theory to brain states defined in terms of connectivity rather than directly in terms of activity. Having identified differences in the temporal dynamics of brain states, our next question was whether differences in the energetic landscape of these states might explain why patients with Parkinson's hallucinations tend to spend more time in a more Segregated state. Network control theory allowed us to specifically test this, by calculating the energy required to be injected into each state in order to move between states, as well as the energy required to maintain each state.

We therefore applied network control theory to the brain states derived from the resting state times series, as has been done in other work, for example, Singleton et al BioRxiv 2021. We used functional connectivity (FC) in a network-based approach as is common in analysing resting state data; ultimately FC is a representation of the relationships between functional activation of different brain regions over a given period of time. So effectively, we are using (dynamic) FC as a way to characterise the brain activation landscape at rest, how this changes in patients with hallucinations and then choose the underlying activity states that we want to examine the transitions between: instead of defining our states of interest by saying “we want activity states where nodes X Y Z have high activity”, we say “we want activity states where nodes X and Y are active or inactive together, and in the opposite direction as Z” (for example). Doing so requires looking at their (time-resolved) statistical relationships: that is, precisely the dynamic FC that we use. To achieve this, we use the sum of connectivity for each region as a representation of that region’s overall FC at that specific timepoint.

Importantly, our primary question here was the relative integration and segregation of the functional connectivity states. Network control theory then allowed us to examine *why* there is a difference in transitioning between these states in one patient group compared with another. Specifically, using this framework, we were able to show that the transition from an Integrated to a Segregated state is energetically more efficient and therefore preferable in patients with PD and visual hallucinations, offering a mechanistic explanation of the observed differences in temporal dynamics.

We have now added a section in the discussion to highlight how the approach here differs from other applications of Network Control Theory and an explanation as above for why we have applied it in this way. We hope that the Reviewer will agree that although this is a relatively novel way of applying network control theory, it is a valid approach to address the question here. We hope that by explicitly stating this and explaining what this approach means in the context of network control theory, that this is now much clearer.

Our changes can be found in the Revised Discussion Page 14, as follows:

“Most studies using network control theory so far have assessed transitions from rest to task³⁴ or activation of a specific functional network³⁵ where brain states are defined as regional activation by selecting which regions should be active or inactive. However, as our key question was to examine the transition between the integrated and segregated states (identified from the dynamic changes in the resting state timeseries from our patients), of dynamic functional connectivity in our patients, such an approach was not straightforwardly applicable in our case. Instead, in the literature on brain-states, it is common to define data-driven brain states in terms of time-resolved patterns of functional connectivity: that is, states are defined in terms of how regions are co-active together over a short period of time, rather than by specifying which regions should be active. Therefore, we used the sum of regional functional connectivity as a summary representation for each previously identified state; in this setting our activity states correspond to brief periods where specific nodes are active or inactive together, rather than nodes having high or low activity per se. In other words, our approach identifies states in terms of nodal co-activation. Although this approach provides insights into the relative ease of each transition, a task vs rest approach would be potentially even more informative and could be examined in future work.”

We also clarified in the Revised Results section, as well as in the Discussion, how the brain states were calculated, so that the different approach used here is more explicit. Our changes are on Pages 6, 7, 8 and 11. As follows:

Results:

“two states of dynamic functional connectivity,”

“Using this framework, the minimal energetic cost to transition from one specific functional brain state (defined as the magnitude of brain activation at a specific time point), and

“For the purposes of this calculation, we represented the Integrated and Segregated states as a vector of summed functional connectivity for each brain region.”

Discussion:

“these two states, each represented as the vector of summed connectivity for each region at each time point”

REVIEWERS' COMMENTS:

Reviewer #2 (Remarks to the Author):

The authors have done a good job in addressing my concerns regarding their brain state definition and discussing their unusual state definition in the limitation section.

To make this – in my opinion critical - aspect of the paper more transparent to readers, I would suggest making an additional minor change to the manuscript:

Page 8:

“For the purposes of this calculation **and in contrast to previous publications**, we represented the Integrated and Segregated states as a vector of sum functional connectivity for each brain region. We then examined whether transition and persistence energies in each state differed between PD-VH and PD-non-VH.”